# TRANSFERABILITY OF COMPOSITIONALITY

## ABSTRACT

Compositional generalization is the algebraic capacity to understand and produce large amount of novel combinations from known components. It is a key element of human intelligence for out-of-distribution generalization. To equip neural networks with such ability, many algorithms have been proposed to extract compositional representations from the training distribution. However, it has not been discussed whether the trained model can still extract such representations in the test distribution. In this paper, we argue that the extraction ability does not transfer naturally, because the extraction network suffers from the divergence of distributions. To address this problem, we propose to use an auxiliary reconstruction network with regularized hidden representations as input, and optimize the representations during inference. The proposed approach significantly improves accuracy, showing more than a 20% absolute increase in various experiments compared with baselines. To our best knowledge, this is the first work to focus on the transferability of compositionality, and it is orthogonal to existing efforts of learning compositional representations in training distribution. We hope this work will help to advance compositional generalization and artificial intelligence research. The code is in supplementary materials.

## 1  INTRODUCTION

Human intelligence (Minsky, 1986; Lake et al., 2017) exhibits compositional generalization, the algebraic capacity to understand and produce large amount of novel combinations from known components (Chomsky, 1957; Montague, 1970). This capacity helps humans to recognize the world efficiently and to be imaginative. It is also beneficial to design machine learning algorithms with compositional generalization skills. Current neural network models, however, generally lack such ability. Compositional generalization is a type of out-of-distribution generalization (Bengio, 2017), where the training and test distributions are different. A sample in such a setting is a combination of several components, and the generalization is enabled by recombining the seen components of the unseen combination during inference. In the image domain, an object is a combination of many parts or properties. In the language domain, a compound word is a combination of multiple words. As an example, we consider two digits are overlapped (Figure 1). Each digit is a component, and it appears in training. A test example has a new combination of two digits.

The main approach for compositional generalization is to learn compositional representations (Bengio, 2013), which contain several component representations. Each of them depends only on the underlying generative factor, and does not change when other factors change. We call this the *compositionality* property, and will formally introduce in Section 3. In the digit example, this means that the representation of one digit does not change when the other digit changes.

Multiple approaches have been proposed to learn compositional representations in the train distribution. However, little discussion has focused on whether the model can still extract the representations in the test distribution. We find that the extraction ability does not transfer naturally, because the extraction network suffers from the divergence of distributions (Bengio, 2017; Pleiss et al., 2020), so that each extracted representation shifts away from the corresponding one in training. Our experiment on the digit example shows that the accuracy drops from 89.6% in training to 49.3% in test (Table 1 in Section 5).

To address the problem, we hope each representation is consistent with the training one while reflecting the test sample. We use an auxiliary network, which has hidden representations as inputs,

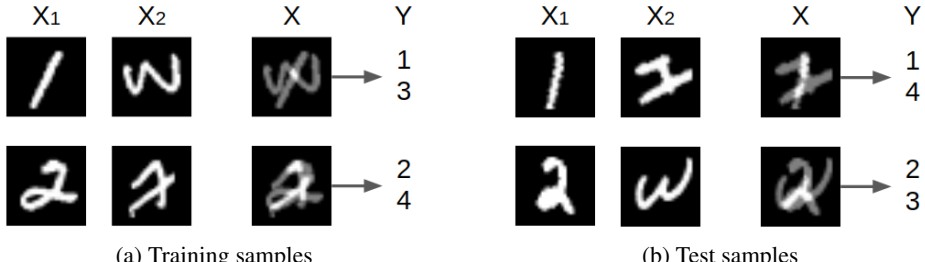

(a) Training samples          (b) Test samples

Figure 1: Examples of compositional generalization with overlapping digits. Each sample is a horizontal block with three images and two digits. The middle image $X$ is input and the right two digits $Y = Y_1, Y_2$ are output. The left two images $X_1, X_2$ are hidden components. $X_1$ is in its original form, and $X_2$ is flipped over left-top to right-bottom diagonal. The sum of the digits is even in train, and odd in test. We hope to learn a prediction model in training, and transfer it to test.

and the original input as output. For a test sample, we regularize each hidden representation in its training manifold, and optimize them to recover the original input. Then we use the optimized representations for prediction. Experimental results show that the proposed approach has more than a 20% absolute increase in various experiments compared to baselines, and even outperforms humans on the overlapping digit task. Our contributions can be summarized as follows.

- We raise and investigate the problem of transferability of compositionality to test distribution. This work is orthogonal to many efforts of learning compositionality in training distribution.

- We propose to address the problem by using an auxiliary reconstruction network with regularized hidden representations as input, and optimize the representations during inference.

- We empirically show that the transferability problem exists and the proposed approach has significant improvements over baselines.

## 2   RELATED WORK

Compositional generalization (Chomsky, 1957; Montague, 1970) is critical in human cognition (Minsky, 1986; Lake et al., 2017; Johnson & et al, 2017; Higgins & et al, 2018; Lake et al., 2019). It helps humans to understand and produce large amount of novel combinations from known components. Broadly speaking, compositional generalization is a type of out-of-distribution (o.o.d.) transferring or generalization, which is also called domain adaptation (Redko et al., 2020) or concept drift (Gama et al., 2014). This is different from traditional i.i.d. setting, where the training and the test distributions are identical. The transferring requires prior knowledge of how the distribution is changed, and compositional generalization has a particular form of such change, as mentioned in the later section.

Compositional generalization is also a desirable property for deep neural networks. Human-level compositional learning (Marcus, 2003; Lake & Baroni, 2018) has been an important open challenge (Yang et al., 2019; Keysers & et al, 2020), although there is a long history of studying compositionality in neural networks. Classic view (Fodor & Pylyshyn, 1988; Marcus, 1998; Fodor & Lepore, 2002) considers conventional neural networks lack systematic compositionality. With the breakthroughs in deep neural networks, there are more contemporary attempts to encode compositionality in deep neural networks. Compositionality in neural networks is actively explored for systematic behaviour (Wong & Wang, 2007; Brakel & Frank, 2009), counting ability (Rodriguez & Wiles, 1998; Weiss et al., 2018) and sensitivity to hierarchical structure (Linzen et al., 2016). Researchers have also proposed multiple related tasks (Lake & Baroni, 2018; Loula et al., 2018; Lake et al., 2019) and methods (Lake et al., 2017; Lake & Baroni, 2018; Loula et al., 2018; Kliegl & Xu, 2018; Li et al., 2019; Lake, 2019; Gorden et al., 2020) for learning compositionality in training distribution. Another line of related work is independent disentangled representation learning (Higgins et al., 2017; Burgess et al., 2018; Kim & Mnih, 2018; Chen et al., 2018; Kumar et al., 2017; Hsieh et al., 2018; Locatello et al., 2019; 2020). Its main assumption is that the expected components are statistically independent in training

data. This setting does not have transferring problem in test, because all combinations have positive joint probabilities in training (please see Section 3).

More recently, there have been approaches for better compositional learning in NLP tasks by elaborating RNN models (Bastings et al., 2018), by using pointer networks (Kliegl & Xu, 2018), or by using two representations of a sentence (Russin et al., 2019; Li et al., 2019). In these tasks, the input can be divided into words, which have consistent information in different distributions. However, such property is not always available, e.g. the overlapping digits. In this paper, we propose to acquire the compositional representations with optimization during inference. We will discuss more in the following sections.

## 3 COMPOSITIONALITY AND TRANSFERABILITY

In this section, we describe compositionality as a model property. We then argue that compositionality may not transfer to test distribution, and discuss ideas of the proposed approach.

**Compositionality** Compositional generalization has different training and test distributions. Samples in both training and test data are combinations of $K$ components. For example, in Figure 1, input $X$ has two digits $X_1, X_2$, and output $Y$ has corresponding labels $Y_1, Y_2$. While a test sample's combination does not appear in training, each component of the test sample appears in training.

A key for compositional generalization is to have compositional representation, which has multiple component representations. Each component representation corresponds to an underlying input component, and it does not change when other components change. We call this property as *compositionality*. A compositional representation is computed from an entangled input, and the extraction network needs to output correct component representations. **If the extraction network transfers to test distribution**, the representations can be correctly extracted in test, so that the compositional generalization is enabled by recombining them.

**Transferability** As mentioned above, to enable compositional generalization, there is an assumption that the property of compositionality should transfer to the test distribution. Since $X_1, \ldots, X_K$ are entangled, the model has the entire $X$ as input. However, input $X$ has different distributions in training and test, so the extraction network suffers from the divergence of distributions (Bengio, 2017; Pleiss et al., 2020). Hence, even if the model is trained to fit the compositionality property in the training distribution, the property is not guaranteed to transfer to the test distribution.

We propose to obtain compositional representations not from the extractor but reversely from an auxiliary network. We extract compositional representations with optimization, and introduce regularization to make each hidden representation in the corresponding training manifold. We then use the optimized hidden representation for prediction. More details are provided in the next section.

## 4 APPROACH

In this section, we introduce the proposed approach from model architecture, training and inference perspectives. The architecture contains three modules, trained with routine end-to-end optimization. The inference, different from conventional procedure, includes three steps: extracts initial hidden representations; optimizes hidden representations as module input; predicts output. Figure 2 contains overall flowcharts, and Algorithm 1 is a summary for the approach. We describe details here.

### 4.1 MODEL ARCHITECTURE

The model takes a sample with input $X$ and label $Y$. We have a representation extractor $g$ with parameter $\phi$, which takes $X$ as input, and outputs $K$ hidden representations $H = H_1, \ldots, H_K$: $H = g(X; \phi)$. We also have a prediction network $f$ with parameter $\theta$, which takes hidden representations $H$ as input, and outputs $\hat{Y}$: $\hat{Y} = f(H; \theta)$. These networks can be some existing networks for compositionality learning. In addition to them, we have an auxiliary network $h$ with parameters $\psi$, which takes hidden representations $H$ as input, and combines them to output $\hat{X}$: $\hat{X} = h(H; \psi)$.

$$X \underset{h(H;\psi)}{\overset{g(X;\phi)}{\rightleftarrows}} H \xrightarrow{f(H;\theta)} Y \qquad X \xrightarrow{g(X;\phi)} H \quad X \underset{h(H;\psi)}{\longleftarrow} H \quad H \xrightarrow{f(H;\theta)} Y$$

(a) Training flowchart. The three modules are trained with end-to-end optimization.

(b) Inference flowchart. (Left) initial hidden representations extraction. (Middle) optimization of hidden representations as module input. (Right) output prediction.

Figure 2: Flowcharts of the proposed approach. $X$ is input, $Y$ is output, and $H$ is hidden representation. The architecture has three modules: $g, h, f$.

---

**Algorithm 1** The proposed approach for training (left) and inference (right). $\alpha, \beta, \gamma, \eta$ are hyper parameters. $K$ is the number of components. $M$ is the number of instances in memory.

Training sample: $X, Y$
1: $H = H_1, \ldots, H_K = g(X; \phi)$
2: $H' = H + \eta\epsilon, \epsilon \in \mathcal{N}(0, I)$
3: $\hat{Y} = f(H'; \theta), \hat{X} = h(H'; \psi)$
4: $\mathcal{L}_{\text{train}} = \text{CE}(Y, \hat{Y}) + \alpha L_2(X, \hat{X}) + \beta L_2(H)$
5: $\hat{\theta}, \hat{\phi}, \hat{\psi} = \arg\min_{\theta, \phi, \psi} \mathcal{L}_{\text{train}}(X, Y, \theta, \phi, \psi)$
6: $\text{Mem}^m = g(X^m; \hat{\phi}), m = 1, \ldots, M$

Inference sample: $X$
1: $H^{\text{init}} = g(X; \hat{\phi})$
2: $\hat{X} = h(H; \hat{\psi})$
3: $\mathcal{L}_{\text{manf}}(H) = \sum_{k=1}^{K} \min_m L_2(H_k, \text{Mem}_k^m)$
4: $\mathcal{L}_{\text{infer}}(X, H) = L_2(X, \hat{X}) + \gamma \mathcal{L}_{\text{manf}}(H)$
5: $\hat{H} = \arg\min_H \mathcal{L}_{\text{infer}}(X, H), H_0 = H^{\text{init}}$
6: $\hat{Y} = f(\hat{H}; \hat{\theta})$

---

### 4.2 TRAINING

In training, we sequentially use the extractor $g$ and predictor $f$, by setting the output of the extractor $H$ as input of the predictor. We have a loss $\mathcal{L}_{\text{original}}$ (containing regularization terms) , such as cross entropy: $\text{CE}(Y, \hat{Y})$, to train a model with compositionality by existing algorithms.

On top of that, we train $\psi$ with $H$ as inputs and $\hat{X}$ as output. We set auxiliary loss as the difference ($L_2$ distance) between $X$ and $\hat{X}$: $\mathcal{L}_{\text{auxiliary}} = L_2(X, \hat{X})$. We also regularize the $L_2$ norm of $H$, $\mathcal{L}_{\text{hidden}} = L_2(H)$, and add noise, $H' = H + \eta\epsilon, \epsilon \in \mathcal{N}(0, I)$, to avoid remembering $X$. $\eta$ is a hyper parameter. The whole train loss $\mathcal{L}_{\text{train}}$ is the combination of the original loss, auxiliary loss $\mathcal{L}_{\text{auxiliary}}$, and regularization $\mathcal{L}_{\text{hidden}}$, with coefficients $\alpha, \beta$.

$$\mathcal{L}_{\text{train}} = \mathcal{L}_{\text{original}} + \alpha\mathcal{L}_{\text{auxiliary}} + \beta\mathcal{L}_{\text{hidden}}$$

We train the model in end-to-end manner. This can be a standard training for neural networks.

$$\hat{\theta}, \hat{\phi}, \hat{\psi} = \arg\min_{\theta, \phi, \psi} \mathcal{L}_{\text{train}}(X, Y, \theta, \phi, \psi)$$

After training, we store hidden representations for $M$ training samples. They are used to restrict test representation manifold during inference.

$$\text{Mem}^m = g(X^m; \hat{\phi}), \quad \forall m = 1, \ldots, M$$

### 4.3 INFERENCE

We use optimization to acquire hidden representations during inference. Given a test sample $X$, the model predicts its output $\hat{Y}$. We use the auxiliary network $h(H; \hat{\psi})$ to search for hidden representations $H$ so that $h$ can output $\hat{X}$ that is close to the original intput $X$. This can be achieved by optimization on $H$ with auxiliary loss $\mathcal{L}_{\text{auxiliary}}$. The initial value $H^{\text{init}}$ is the output of extractor $g$: $H^{\text{init}} = g(X; \hat{\phi})$.

We also add a manifold regularization term $\mathcal{L}_{\text{manf}}$ to constrain each hidden representation to lie in the corresponding training manifold. For a test sample, we compute the minimum of $L_2$ distance between each of its hidden representation $H_k$ and the corresponding representations in memory $\text{Mem}_k^m$. We

then use the sum of the distances as regularization.

$$\mathcal{L}_{\text{manf}}(H) = \sum_{k=1}^{K} \min_{m=1,\dots,M} L_2(H_k, \text{Mem}_k^m)$$

Inference loss $\mathcal{L}_{\text{infer}}$ is the combination of auxiliary loss and the regularization, with $\gamma$ as a coefficient.

$$\mathcal{L}_{\text{infer}}(X, H) = \mathcal{L}_{\text{auxiliary}}(X, \hat{X}) + \gamma \mathcal{L}_{\text{manf}}(H)$$

Then, we obtain the hidden representations by optimization.

$$\hat{H} = \arg\min_{H} \mathcal{L}_{\text{infer}}(X, H), \quad H_0 = H^{\text{init}}$$

We get prediction from the optimized hidden representations: $\hat{Y} = f(\hat{H}; \hat{\theta})$.

## 5  EXPERIMENTS

In this section, we show examples that, given a model with compositionality in training distribution, the compositionality does not transfer to test distributions, and how the proposed approach is applied to these cases. As our focus is orthogonal with learning compositional model in training distribution, we obtain this model by directly providing true labels for each compositional component during training. Then we evaluate the transferability to test distribution.

Since this is the first work for the transferability problem of compositionality, we do not have previous baselines, so that we use standard deep neural network models as baselines, and also compare with variations of the proposed approach. The main changes of the proposed approach include using noise, auxiliary network and manifold regularization. A variation removes one of these changes, so it is also an ablation experiment. The details of hyper parameters and experiment settings can be found in Appendix A. For all the experiments, we use accuracy as metric. A prediction is correct if and only if all the components are correctly predicted. We repeat each experiment five times and report mean and variance.

| Method | Overlapping digits Test dist. | Overlapping digits Train dist. | Compound words Test dist. | Compound words Train dist. | Colored digits Test dist. | Colored digits Train dist. |
|---|---|---|---|---|---|---|
| Human | $10.0 \pm 7.7$ | - | - | - | - | - |
| Standard DNN | $49.3 \pm 0.9$ | $89.6 \pm 0.3$ | $27.0 \pm 6.7$ | $100.0 \pm 0.0$ | $48.8 \pm 4.3$ | $99.0 \pm 0.2$ |
| Variation -noise | $44.2 \pm 0.9$ | $79.8 \pm 0.9$ | $11.3 \pm 6.2$ | $100.0 \pm 0.0$ | $37.7 \pm 4.0$ | $96.9 \pm 0.6$ |
| Variation -auxiliary | $51.1 \pm 0.7$ | $88.7 \pm 1.1$ | $46.8 \pm 3.7$ | $100.0 \pm 0.0$ | $10.6 \pm 3.5$ | $98.7 \pm 0.3$ |
| Variation -manifold | $60.8 \pm 3.7$ | $69.8 \pm 2.3$ | $\mathbf{51.9 \pm 3.3}$ | $99.8 \pm 0.5$ | $81.8 \pm 1.4$ | $92.4 \pm 1.0$ |
| Proposed | $\mathbf{69.4 \pm 0.3}$ | $81.1 \pm 1.0$ | $51.2 \pm 3.0$ | $100.0 \pm 0.0$ | $\mathbf{91.2 \pm 0.6}$ | $96.2 \pm 0.4$ |

Table 1: Evaluation accuracy (%). We see that the proposed method has significant improvement over the standard DNN on three datasets and outperforms human on overlapping digit dataset.

### 5.1  EXPERIMENTS ON OVERLAPPING DIGITS

The first experiment is on overlapping hand written digit recognition, as shown in Figure 1. We construct the dataset from MNIST (LeCun et al., 1998). A sample is made by overlapping and taking the average of two original images, the first one in its original form, and the second flipped over up-left to down-right diagonal. The output is a vector of the two labels $Y = Y_1, Y_2$ (not exchangeable). Each original label has 10 possible values, so the output has 100 possible values. To evaluate compositional generalization, we use different distributions in training and test. In train, the sum of the two labels is even, i.e. $(Y_1 + Y_2) \bmod 2 = 0$. In test, the sum is odd.

As a baseline, we use a standard neural network with two sub networks, each for an output. Each of the network is a three layer convolutional neural network. We train the model with cross entropy on both outputs.

The proposed method uses an auxiliary network that takes hidden layer as input and outputs the reconstruction of the original input. The auxiliary network has two sub-networks each with one

| Input | Outputs | Hidden | |
|---|---|---|---|
| januarymarch | 0, 2 | january | march |
| februaryoctober | 1, 9 | february | october |
| januaryfebruary | 0, 1 | january | february |
| augustmay | 7, 4 | august | may |

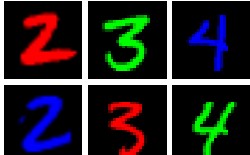

Table 2: Examples of compound word experiments. The output labels align with corresponding hidden words. Upper is train, lower is test.

Figure 3: Examples of input for colored digits experiments. Output is the digit label and color label. Upper is train, lower is test.

hidden representation as input. Each sub-network is a three layer trans-convolutional neural network, and we average the outputs to recover the original input. We use $L_2$ loss as the auxiliary loss.

We also collect human performance data through crowd sourcing. There are 27 participants, and each person works on 20 fixed samples randomly selected from test data. Please refer to Appendix B for more details.

The results in Table 1 (left) show that the proposed method has significant improvement over the baseline by about 20% absolute increase, and it also outperforms humans. The ablation study shows that performances drop in the experiments, indicating that all the modifications in the proposed approach are necessary to achieve the result.

## 5.2 EXPERIMENTS ON COMPOUND WORDS

We also conduct experiments for language processing. Language has natural units of words, which have consistent information across different distributions. So we design a setting that we cannot use this property. We consider a problem that converts a compound word to two words. We construct compound words from two month names (January to October), e.g. "julyfebruary". The output label is the zero-based index of the month (0 for January). We have each character as an input unit, and assign a one-hot representation to it. Other problem settings are the same as previous one. Please see Table 2 for more examples.

In baseline, We use two feed forward neural networks, each for an output. For the auxiliary network, we also use two feed forward neural networks, and average their outputs. Each feed forward network has three hidden layers. The training and other settings for baseline, proposed approach and ablations are the same as the previous experiment.

The results listed in Table 1 (middle) demonstrate that the proposed method is significantly better than the baseline by around 24% absolute increase. We find removing manifold regularization in ablation study shows slight improvement over the proposed approach. Other ablation experiments have significant reductions. This might be because regularizing manifold is not important when inputs (characters) are discrete.

## 5.3 EXPERIMENTS ON COLORED DIGITS

We also explore the capability of the proposed approach to another hand written digit problem with different types of components: digits and colors. We construct the dataset from MNIST (LeCun et al., 1998) by changing the color of digits. We use digit label $Y_1$ as a component (0-9), and color label $Y_2$ as another (0-2). Color label is 0, 1, 2 for red, blue, green, respectively. In training, we use label combinations with: $Y_1 \bmod 3 \neq Y_2$. In test, we use the rest of the combinations. Please refer to Figure 3 for examples.

We use two compositional neural networks for each output, respectively. Each network has three hidden layers. For the auxiliary network, we concatenated the hidden representation as input, and use a three layer trans-convolutional neural network. Other settings for the methods and ablation are the same as the overlapping digit experiment.

The results shown in Table 1 (right) demonstrate that the proposed method is significantly better than the baseline by more than 40% absolute increase. It also outperforms the ablations, indicating

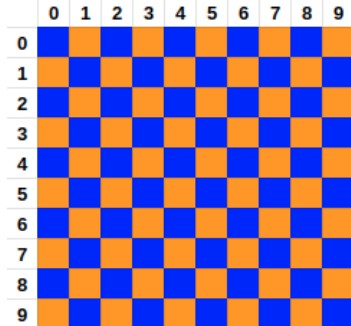

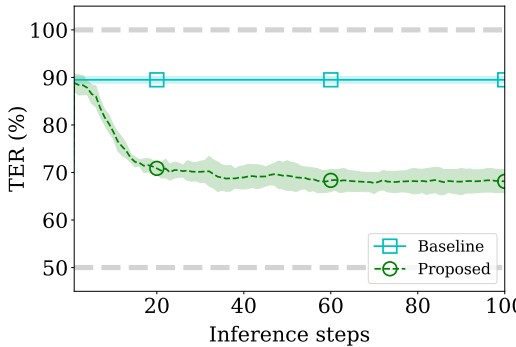

Figure 4: Out-of-distribution problem has different distributions in training and test. We hope to learn a model in training distribution (blue), and use it in test distribution (orange).

Figure 5: Transfer Error Rate (TER) during inference optimization. 100% is upper bound by definition. 50% means errors are balanced for in-distribution and out-of-distribution.

that all the modifications are necessary. Among them, auxiliary network contributes the most to the performance improvement.

## 6 DISCUSSIONS

In this section, we perform visualization and error analysis to better understand the experimental results and the behavior of the proposed approach.

### 6.1 DISTRIBUTION VISUALIZATION

We visualize hidden representations of both baseline and the proposed approach for overlapping digit experiments (Figure 6). We use t-SNE (Maaten & Hinton, 2008) to reduce each of two hidden representations to one dimension, and jointly plot them along horizontal and vertical coordinates, respectively. Training samples are blue, and test samples are orange.

Our expectation is a chess board like distribution, similar to the true underlying distribution (Figure 4). Note that though there are 10 labels, the expected results may not be $10 \times 10$ colored blocks, because the labels, along with the representation, may not be in order (e.g. switching label 5, 6 reduces two blocks lines), and the first and the second hidden representations may differ for the same digit.

We find the visualization of the proposed approach (Figure 6b) is closer to the expectation than the baseline one (Figure 6a). The proposed approach has less empty areas, indicating that it can recombine the components in the out-of-distribution setting. This analysis demonstrates that the proposed approach works in the expected way.

### 6.2 SAMPLE VISUALIZATION

We also hope to visualize concrete samples for the proposed approach. Since we have the auxiliary network with two sub-networks for each hidden component representation, we visualize their outputs, and compare with the ground-truth (Figure 7). The result shows that the original input and hidden components are reasonably recovered for both training and test samples. This means that the proposed approach is able to extract information for each component in training distribution, and transfer the ability in test distribution.

### 6.3 TRANSFER ERROR ANALYSIS

We analyze errors to show that the inference optimization process helps addressing the transfer problem. When the model makes mistakes in test, the predicted output may correspond to in-distribution or out-of-distribution label pairs. We investigate how frequent errors are associated with the distribution transfer. We define a metric of Transfer Error Rate (TER) to measure this. It is the

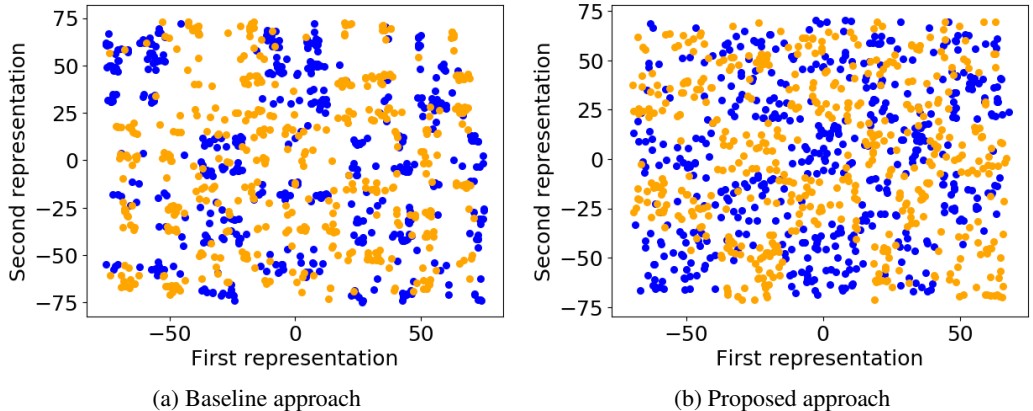

(a) Baseline approach          (b) Proposed approach

Figure 6: Visualization of hidden representations. Each representation is reduced to one dimension via t-SNE (Maaten & Hinton, 2008). We plot them jointly, training in blue, and test in orange. The proposed approach (b) is close to expected chess board like result, similar to Figure 4.

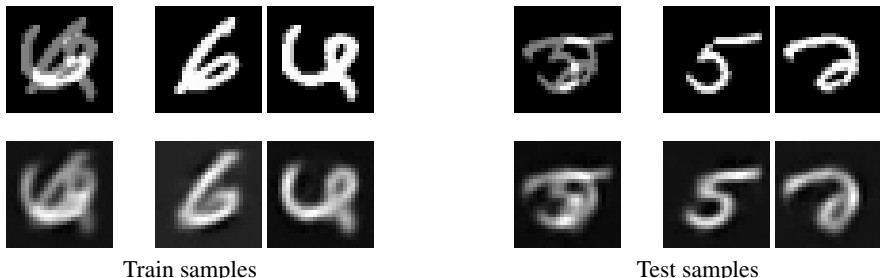

Train samples          Test samples

Figure 7: Visualization for the proposed approach. The first row is ground truth. The second row is recovered images from auxiliary network. The first column is the overlapping image. The second and third columns are the images for the first and the second components, respectively. The results show that the proposed approach is able to learn and transfer the ability to extract correct components.

number of errors predicted to be in wrong distribution, over all errors. In this setting, the sum of labels is odd in test, and even in training, so TER is the number of errors with even predicted label sum over all errors. If this rate is high (100% is upper bound by definition), it means that most of errors are associated with transfer. If it is around 50%, it means that the errors are balanced in distributions.

TER is 89.5±0.7% for baseline, and 68.3±2.8% for the proposed approach, with 500 test samples. Also, before the inference optimization in the proposed approach, the value is 88.8±2.0%. The results show that the baseline has most of the errors to be transfer error, and the propose approach reduces it significantly. Also, the reduction is during optimization in inference, because the value is close to baseline before optimization. This indicates that the proposed approach is effective to reduce TER, and optimization is an important factor for it. Please refer to Figure 5 for more details.

## 7 CONCLUSION

In this paper, we discuss our finding that compositionality does not transfer naturally from training to test data distributions. We further propose to address this problem with an auxiliary reconstruction network and a regularized optimization procedure during the inference stage. Experimental results show that the proposed approach has more than 20% absolute increase in various experiments comparing with baselines, and even outperforms human on the overlapping digit recognition task. We hope this work would reshape our thoughts on transferability of compositionality and help to advance compositional generalization and artificial intelligence research.

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

## A  EXPERIMENT SETTINGS

### A.1  EXPERIMENTS ON OVERLAPPING DIGITS

The data of overlapping digits experiment are generated from MNIST dataset (LeCun et al., 1998), which contains 60,000 training data and 10,000 test data. We use the original training data in training and the original test data in test. In training, we use the combinations with even label sum. In test, we use the combinations with odd label sum (evaluation on test distribution). As a reference, we also use the combinations with even label sum (evaluation on training distribution).

Each input image has a shape of $28 \times 28 \times 3$ for height, width and depth. Extraction network $g(\cdot; \phi)$ has two convolutional neural sub-networks. Each of them has three convolutional layers. The first convolutional layer has $3 \times 3$ kernel and $1 \times 1$ strides with 32 depth and ReLU activation. It is followed by a max pooling layer with $2 \times 2$ kernel and $1 \times 1$ strides. The second convolutional layer has $3 \times 3$ kernel and $1 \times 1$ strides with 64 depth and ReLU activation. Then the representation is flatten and fed to a fully connected neural network with 32 hidden nodes and linear activation. Prediction network $f(\cdot; \theta)$ has a single layer fully connected neural network with Softmax activation for each pair of hidden representation and output. Auxiliary network $h(\cdot; \psi)$ has two trans-convolutional neural networks. A hidden representation is converted to $7 \times 7 \times 32$ representation (width, height, depth) by a fully connected neural network with ReLU activation. The first trans-convolutional layer has $3 \times 3$ kernel and $2 \times 2$ strides with 64 depth, SAME padding and ReLU activation. The second trans-convolutional layer has $3 \times 3$ kernel and $2 \times 2$ strides with 32 depth, SAME padding and ReLU activation. The third trans-convolutional layer has $3 \times 3$ kernel and $1 \times 1$ strides with 1 depth, SAME padding and linear activation.

For the hyper parameters, we use $\alpha = 1$, $\beta = 0.3$, $\gamma = 0.03$, $\eta = 1$, $M = 500$. In training optimization, we use Adam optimizer (Kingma & Ba, 2014) for 1,000 steps with batch size 500 and learning rate 0.001. In inference optimization, we also use Adam optimizer for 100 steps with batch size 500 and learning rate 0.3. We find the memory size $M$ does not change the result significantly. We have 69.0±0.6% for $M = 100$, 68.2±1.0% for 300, 69.4±0.3% for 500 (reported), 69.6±0.5% for 700 and 69.9±0.4% for 900 (Figure 8).

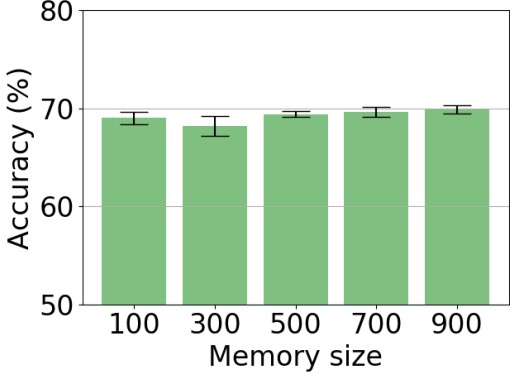

Figure 8: Evaluation accuracy for different memory sizes in the proposed method.

### A.2  EXPERIMENTS ON COMPOUND WORDS

In the compound words experiment, we have inputs with different lengths. We first patch an additional character symbol to make all inputs the same length of 18. Then, we convert each character to a one-hot representation with size 27. we use three layer fully connected neural networks as sub-networks for extractor. The input is flatten and fed to each sub-network. The first layer has 96 hidden nodes with ReLU activation. The second layer has 64 hidden nodes with ReLU activation. The third layer has 32 hidden nodes with linear activation. Prediction network is a single layer fully connected network with Softmax activation for each pair of hidden representation and output. The auxiliary network also has two three layer fully connected sub-networks. The first layer has 64 hidden nodes

with ReLU activation. The second layer has 96 hidden nodes with ReLU activation. The third layer has 486 hidden nodes with linear activation. Then the nodes are reshaped to $18 \times 27$.

For the hyper parameters, we use $\alpha = 1$, $\beta = 0.1$, $\gamma = 0.01$, $\eta = 1$, $M = 500$. In training optimization, we use Adam optimizer for 1,000 steps with batch size 500 and learning rate 0.001. In inference optimization, we also use Adam optimizer for 100 steps with batch size 500 and learning rate 0.3.

### A.3 EXPERIMENTS ON COLORED DIGITS

The data for colored digits experiment are created from MNIST dataset. Each input image is in shape of $28 \times 28 \times 3$ for height, width and depth. The extractor has two sub-networks, each has three convolutional layers. The first convolutional layer has $3 \times 3$ kernel and $1 \times 1$ strides with 32 depth and ReLU activation. A max pooling layer follows it with $2 \times 2$ kernel and $1 \times 1$ strides. The second convolutional layer has $3 \times 3$ kernel and $1 \times 1$ strides with 64 depth and ReLU activation. Then the representation is flatten and fed to a fully connected neural network with 32 hidden nodes with linear activation. Prediction network has a fully connected network for each pair of hidden representation and output with Softmax activation. For auxiliary network, we first concatenate the hidden representations to have 64 nodes. It is then converted to $7 \times 7 \times 32$ representation (width, height, depth) by a fully connected neural network with ReLU activation. The first trans-convolutional layer has $3 \times 3$ kernel and $2 \times 2$ strides with 64 depth, SAME padding and ReLU activation. The second trans-convolutional layer has $3 \times 3$ kernel and $2 \times 2$ strides with 32 depth, SAME padding and ReLU activation. The third trans-convolutional layer has $3 \times 3$ kernel and $1 \times 1$ strides with 3 depth, SAME padding and linear activation.

For the hyper parameters, we use $\alpha = 1$, $\beta = 0.3$, $\gamma = 0.03$, $\eta = 1$, $M = 500$. In training optimization, we use Adam optimizer for 1,000 steps with batch size 500 and learning rate 0.001. In inference optimization, we also use Adam optimizer for 100 steps with batch size 500 and learning rate 0.3.

## B CROWD SOURCING FOR HUMAN PERFORMANCE

We collect data on human performance through crowd-sourced survey. The survey has training and test sections. We select 20 images uniformly at random from the training distribution as training data. We also select 20 images uniformly at random from the test distribution as test data (Figure 9). In the training section we state that each picture is an overlap of two pictures containing two numbers. Then we ask survey takers to discover the rule through the training examples before they answer questions. In the test section we put up test images, and two multiple-selection (0-9) list boxes under each image. Then we ask the survey takers to identify the correct answers. In this survey we had 27 responses. The mean test accuracy is 10.0% with standard deviation of 7.7%.

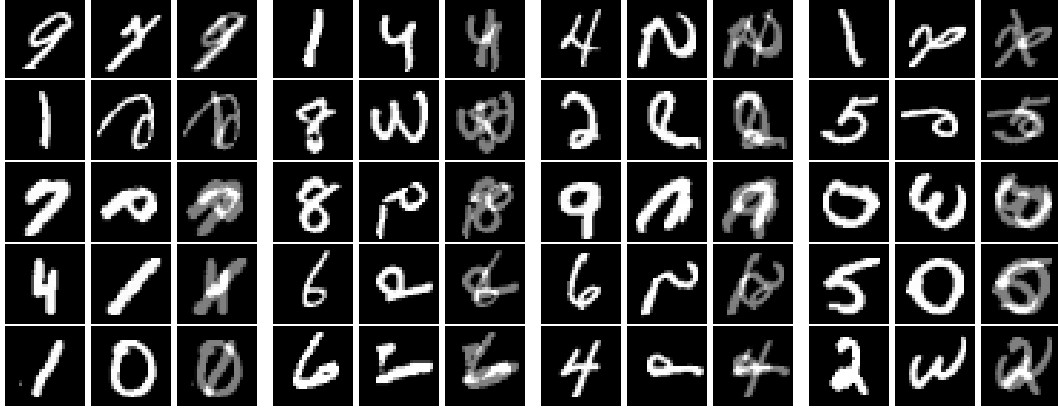

Figure 9: Test samples and corresponding component images for crowd sourcing. Each sample is a horizontal block with three images. The left two images are hidden components. The right image is the overlapping of them. Only the overlapping image is shown to participants.

