# OpenReview forum: "Transferability of Compositionality"
_ICLR.cc/2021/Conference — Reject_

### Official Review · AnonReviewer4 · 2020-10-21
**Paper about a relevant topic, but with insufficient motivation and grounding in previous work**

**Rating:** 3
**Confidence:** 5

**Review:**

*Summary*

This paper proposes an architecture that addresses transferability of compositionality. The proposed architecture consists of three components: a network that transforms the input X into a series of hidden representations {H_1, H_2, ... H_K}, a network that reconstructs the input X from this series of hidden representations, and a prediction network that generates a prediction from the hidden representations. The authors propose several datasets meant to address transferability of compositional generalisation, and show that their architecture significantly improves standard DNN architectures as well as humans on these datasets.

*Motivation for score*

Compositional generalization in neural networks is a relevant and hot topic, with still many open questions. This paper aims to contribute on this topic and proposes some interesting datasets. However, However, despite citing several papers addressing compositionality in neural networks in the related work section, I am not convinced that the authors have properly understood the questions that are asked in this domain and were able to address them properly. Below, I outline my concerns.

1. Definition of compositionality

I do not find the definition of compositionality that the authors propose well motivated.
- None of the three papers cited in the introduction to motivate the work actually has the word "compositionality" in the paper
- The authors claim that previous work has focused just on whether models can extract compositional representations in the training distribution while ignoring the test distribution, while actually most recent papers they cite in related work test compositionality by considering very specific train/test splits
- The author's definition of compositional generalisation does not seem to take into account that compositionality is traditionally a property of mapping between input and output, not of a model itself. In addition to that, whether the mapping between input and output is compositional does not depend on what is in the train and what is in the test distribution. A model can understand the compositional structure of a dataset also if it has been trained on *all* examples of the dataset, only it will be impossible to behaviourally evaluate if it has. For this reason, much previous work on compositionality in neural networks has created datasets where the training and testing data were distributionally different (as also the authors of this paper do).
I would recommend the authors to have a look at the paper _Compositionality decomposed: how do neural networks generalise?_ (Hupkes et al.; 2020), for a detailed account of compositionality in the context of neural networks. In particular, their section on _localism_ is particularly important for the author's definition of compositionality.

2. Architecture

I find the proposed architecture interesting, but it is not completely clear to me how it differs from an auto-encoder setup where the encoding is larger than the input instead of smaller (it is very well possible I misunderstood). Nevertheless, it can be interesting to see if auto-encoding based architectures behave better on datasets proposed to evaluate compositionality. One thing that is not clear to me is how the number of components _K_ is determined.

3. Data

I appreciate the effort of the authors to design new datasets that test out-of-distribution generalisation. I do have a few comments/questions:
- If the main motivation for wanting compositional generalisation is that this is an important capacity of humans, isn't it a problem that humans perform very poorly on the dataset (much worse than the best deep neural network)?
- What is the motivation for using a new dataset, rather than one of the previously proposed datasets for out-of-distribution generalisation?
- It is nice that the authors try to include tests from different domains, but I think that calling a dataset mapping inputs like "januarymarch" to (0, 2) cannot really be called "natural language processing"

Overall, I do not believe that this paper should be accepted for the conference.

_**Update after author response:** I have read the author response, but do not find that the answers really address my concerns.  I have also not really seen any improvements in the paper itself._

---

> ### Author Response · Authors · 2020-11-16
> **Reply to Reviewer 4**
>
> Thank you for helping us to improve the paper.
>
> Q1: I do not find the definition of compositionality that the authors propose well motivated.
> None of the three papers cited in the introduction to motivate the work actually has the word "compositionality" in the paper
>
> A1: It is mainly motivated from Lake et. al. 2017.
>
> Q2: The authors claim that previous work has focused just on whether models can extract compositional representations in the training distribution while ignoring the test distribution, while actually most recent papers they cite in related work test compositionality by considering very specific train/test splits
>
> A2: The previous work either uses language examples where the transferring problem is not prominent, or does not measure compositional generalization.
>
> Q3: The author's definition of compositional generalisation does not seem to take into account that compositionality is traditionally a property of mapping between input and output, not of a model itself.
>
> A3: We consider the input to output compositional generalization approach with compositional hidden representations.
>
> Q4: I would recommend the authors to have a look at the paper Compositionality decomposed: how do neural networks generalise? (Hupkes et al.; 2020)
>
> A4: Thank you. We will look into it.
>
> Q5: I find the proposed architecture interesting, but it is not completely clear to me how it differs from an auto-encoder setup where the encoding is larger than the input instead of smaller (it is very well possible I misunderstood). Nevertheless, it can be interesting to see if auto-encoding based architectures behave better on datasets proposed to evaluate compositionality.
>
> A5: Auto-encoding is for unsupervised problems, and we consider general supervised learning with different input and output. We will investigate how to compare with auto-encoding methods.
>
> Q6: One thing that is not clear to me is how the number of components K is determined.
>
> A6: K is given as prior knowledge in this case, and we consider this as a part of the training problem. We focus on the transfer problem in inference.
>
> Q7: If the main motivation for wanting compositional generalisation is that this is an important capacity of humans, isn't it a problem that humans perform very poorly on the dataset (much worse than the best deep neural network)?
>
> A7: The main motivation is that capability of compositional generalization itself. The human performance shows the task is a difficult one.
>
> Q8: What is the motivation for using a new dataset, rather than one of the previously proposed datasets for out-of-distribution generalisation?
>
> A8: Previous dataset for out-of-distribution generalization, such as SCAN dataset, have word unit based language tasks where the transferring problem is not prominent.
>
> Q9: It is nice that the authors try to include tests from different domains, but I think that calling a dataset mapping inputs like "januarymarch" to (0, 2) cannot really be called "natural language processing"
>
> A9: We would modify this as a language problem.

---

### Official Review · AnonReviewer1 · 2020-10-27
**the paper considers an interesting general problem, but the concrete supervised learning instantiation is problematic**

**Rating:** 4
**Confidence:** 3

**Review:**

The paper introduces a “transferability of compositionality” problem and proposes an approach to alleviate it. The said problem may arise when one trains neural models to produce “compositional” representations of the input. In the paper “compositional representations” consist of multiple vectors which are supposed to correspond to semantically meaningful aspects of the input, for example different objects in the case of images or different parts of compound words in the case of linguistic inputs. The transferability problem arises when there is a difference between training and test distributions, namely when certain combinations of objects have different probabilities in training & testing. The proposed solution at inference time is to project object representations to the manifold of individual object representations. The manifold is estimated by saving representations of individual object representations from the training time.

The problem that the paper considers is an interesting one. There have been a lot of papers on learning object-oriented representations recently [1, 2], and an implicit assumption in all these works is that there is no statistical dependency between which objects that occur in the scenes. There is also the literature on disentangled representations that the paper extensively cites, where the independence assumption is also common.

My concerns regarding the paper are as follows:
- Positioning with the respect to the prior work. The literature on learning object-oriented representations is not cited. The work on disentangled representation is cited, but still new setups are created from scratch.
- Related to the previous point, the use of full supervision (in the form of labels) in the proposed tasks strikes me as a deviation from most previously used setups. Previous work aimed to learn compositional representations without supervision, often positioning their efforts as a cog-sci-style inquiry in building human-like models. The use of labels makes this look less like a cog-sci and more like a machine learning paper. Viewing the work as an ML paper, one thing that stands out is the lack of connections to any applied ML problem.
- I think the negative results in the paper would look stronger if pretrained image- and language- processing models were used in all experiments (e.g. Contrastive Predicting Coding & BERT)

The proposed method seems appropriate for the tasks that the paper considers. The experiments appear to be technically sound. My main concerns are thus focused on the motivation of the proposed tasks themselves and the positioning with respect to their prior work. I think a great direction to improve the paper would be to add experiments without supervised learning and using 3D-rendered images with multiple objects as it is done e.g. in [1] and [2].

Few comments on writing:
- Algorithm 1 is very confusing because sample-level steps 1-4 are mixed with dataset-level steps 5 and 6.
- A confusing sentence in the intro: “For a test sample, we regularize each hidden representation in its training manifold, and optimize them to recover the original input”
- for colored digits experiments you might want to compare to and cite [3]

- [1] “Multi-Object Representation Learning with Iterative Variational Inference” by Greff et al, 2020
- [2] “MONet: Unsupervised Scene Decomposition and Representation” by Burgess et al, 2019
- [3] “Invariant Risk Minimization” by Arjovsky et al, 2019

---

> ### Author Response · Authors · 2020-11-16
> **Reply to Reviewer 1**
>
> Thank you for the constructive suggestions.
>
> Q1: Positioning with the respect to the prior work. The literature on learning object-oriented representations is not cited. The work on disentangled representation is cited, but still new setups are created from scratch.
>
> A1: We will cite the object-oriented representation work. The disentangled representation setup does not consider compositional generalization.
>
> Q2: Related to the previous point, the use of full supervision (in the form of labels) in the proposed tasks strikes me as a deviation from most previously used setups.
>
> A2:  We use supervision to focus on the transfer problem during inference.
>
> Q3: Viewing the work as an ML paper, one thing that stands out is the lack of connections to any applied ML problem.
>
> A3: This paper focuses on discussing the problem of transferability and finding fundamental mechanisms to address it. We do not extend to applications with compositional generalization here.
>
> Q4: I think the negative results in the paper would look stronger if pretrained image- and language- processing models were used in all experiments (e.g. Contrastive Predicting Coding & BERT) The proposed method seems appropriate for the tasks that the paper considers.
>
> A4: Thank you for the suggestion. We will look into it.
>
> Q5: My main concerns are thus focused on the motivation of the proposed tasks themselves and the positioning with respect to their prior work. I think a great direction to improve the paper would be to add experiments without supervised learning and using 3D-rendered images with multiple objects as it is done e.g. in [1] and [2].
>
> A5: We will investigate it.
>
> Q6: Algorithm 1 is very confusing because sample-level steps 1-4 are mixed with dataset-level steps 5 and 6.
>
> A6: Thank you for mentioning. We will clarify.
>
> Q7: A confusing sentence in the intro: “For a test sample, we regularize each hidden representation in its training manifold, and optimize them to recover the original input”
>
> A7: This sentence is the motivation of the approach, addressing the previous sentence “each extracted representation shifts away from the corresponding one in training”.
>
> Q8: for colored digits experiments you might want to compare to and cite [3]
>
> A8: Thank you, we will find more details.

---

### Official Review · AnonReviewer3 · 2020-10-29
**Review of Transferability of Compositionality**

**Rating:** 3
**Confidence:** 3

**Review:**

## Summary

This paper studies "compositionality" and in particular the way in which it "transfers" on test data. They run simple baselines on three experiments (overlapped MNIST, colored MNIST and concatenated month names) and find that the baselines do not learn compositional representation. They proposed the use of an *auxiliary reconstruction network and a regularized optimization* which improves on these baselines.

## Analysis

The authors frequently say that *compositionality may not transfer to test distribution* but I have a hard time understanding exactly what they mean by this. As I understand it, "compositionality" is a property of a representation. Do authors mean that, on the test data, the representation of an input is able to separate multiple components, yet the same network does not separate the components on the test data? It may be true for the models they trained here, but I would have appreciated a comparison with other methods. As such, I find the claims of this paper difficult to evaluate with respect to previous work. They claim theirs is the *first work for the transferability problem of compositionality* which I find really hard to believe. I would have appreciated a thorough study of the "compositionaly" limitations of previous techniques.

I found section 4 particularly hard to understand. A lot of symbols, equations and nomenclatures seem to be used with too little introduction. As a result, I cannot vouch for the correctness of this section.

Given their claim that this is the *first work for the transferability problem of compositionality* the experiments presented on section 5 are on a new dataset and are not compared to previous work. Moreover, the proposed experiments seem relatively simple (two overlapped MNIST digits, colored MNIST digits, concatenated month names) and their baseline seem trivial (*we use a standard neural network with two sub networks, each for an output*)

## Conclusion

Overall, I find the claim of this paper substantial, while the experiments are relatively simple with trivial baselines and an absence of comparison to related work.

## Typos

I find the text difficult to read, it would benefit from a thorough revision. Ex: *This work is orthogonal to many efforts of learning compositionality in training distribution.*

---

> ### Author Response · Authors · 2020-11-16
> **Reply to Reviewer 3**
>
> Thank you for the comments.
>
> Q1: The authors frequently say that compositionality may not transfer to test distribution but I have a hard time understanding exactly what they mean by this. As I understand it, "compositionality" is a property of a representation. Do authors mean that, on the test data, the representation of an input is able to separate multiple components, yet the same network does not separate the components on the test data? It may be true for the models they trained here, but I would have appreciated a comparison with other methods. As such, I find the claims of this paper difficult to evaluate with respect to previous work. They claim theirs is the first work for the transferability problem of compositionality which I find really hard to believe. I would have appreciated a thorough study of the "compositionaly" limitations of previous techniques.
>
> A1: Your understanding of compositionality is correct. We focus on the case to use compositional representation to achieve compositional generalization, and this is the first work to focus on the transferability in such cases.
>
> Q2: I found section 4 particularly hard to understand. A lot of symbols, equations and nomenclatures seem to be used with too little introduction. As a result, I cannot vouch for the correctness of this section.
>
> A2: We will make them more clear.
>
> Q3: Given their claim that this is the first work for the transferability problem of compositionality the experiments presented on section 5 are on a new dataset and are not compared to previous work. Moreover, the proposed experiments seem relatively simple (two overlapped MNIST digits, colored MNIST digits, concatenated month names) and their baseline seem trivial (we use a standard neural network with two sub networks, each for an output)
>
> A3: We focus on the compositional generalization with compositional representation, where the works are still in the stage of using illustrative examples to find the fundamental mechanisms.
>
> Q4: Typos. I find the text difficult to read, it would benefit from a thorough revision.
>
> A4: We will make them more clear.

---

> > ### Comment · AnonReviewer3 · 2020-11-17
> > **Thanks for your reply**
> >
> > Dear authors,
> >
> > First, thanks a lot for your answers.
> >
> > In my original comment I wrote: "It may be true for the models they trained here, but I would have appreciated a comparison with other methods. As such, I find the claims of this paper difficult to evaluate with respect to previous work. [...] I would have appreciated a thorough study of the "compositionaly" limitations of previous techniques."
> >
> > Is there anything you can say about this? My biggest challenge is that I find it hard to believe that representations behave significantly differently on test data yet is able to perform the downstream task with similar accuracy.
> >
> > Also, I will be happy to look at a revised version of the manuscript to see if I can get a better understanding of section 4.

---

> > > ### Author Response · Authors · 2020-11-23
> > > **Reply to Reviewer 3**
> > >
> > > Thank you for the question.
> > >
> > > A: Here, we just show there exist transferring problems in some compositional generalization tasks, but the problems may not appear in other settings. Also, in previous work with downstream tasks, the focus is on disentangled representation learning. It is likely that the training dataset has marginal independence between underlying factors. So the settings are quite different.

---

### Official Review · AnonReviewer2 · 2020-11-03
**The submission claims to be the first to truly test compositional generalization (ignoring all prior work that does) and fails to motivate an algorithm (whose relation to prior methods is not discussed) applied to toyish datasets (whose relations to existing evaluations that investigate compositionality are not discussed).**

**Rating:** 2
**Confidence:** 5

**Review:**

##### Impact:

The submission claims that other works that investigation compositionality in representation learning do not actually test compositional generalization ("because all combinations have positive joint probabilities in training"). However, I disagree that this is the case in prior work; here are some examples of prior works that correctly hold out novel combinations (of underlying components) for test time:
- https://openreview.net/forum?id=HJz05o0qK7
- http://papers.nips.cc/paper/8825-learning-by-abstraction-the-neural-state-machine
- https://arxiv.org/abs/1910.09113
- https://arxiv.org/abs/1912.09713
- https://arxiv.org/abs/1912.12179

There are many such examples; they are too numerous to list here.

##### Quality:

The algorithmic components in Section 4 are not adequately motivated, and the relationship of the algorithm to prior work in compositional representation learning is not discussed.

  The evaluation tasks are extremely simple (overlayed MNIST digits and conjoined word token) and are, as such, far from the complexity of existing work on compositionality (which can deal with, for example, naturalistic image data; see the references above for examples of such works).

##### Clarity:
There are many points of ill clarity / inconsistencies; for example:
- "The main approach for compositional generalization is to learn compositional representations" Is this really the "main approach"? Compositional generalization has been studied in many contexts outside of representation learning (e.g., see https://semanticsarchive.net/Archive/jcyZDc1Y/Goldberg.Compositionality.RoutledgeHandbook.pdf)

- "We find that the extraction ability does not transfer naturally, because the extraction network suffers from the divergence of distributions" Why is it assumed here that there is an extraction network? The "extraction network" is referred to several times in the introduction and methods section prior to its introduction/explanation.

- "compositional generalization is a type of out-of-distribution (o.o.d.) transferring or generalization, which is also called domain adaptation" This is inconsistent with the previously discussed definition of compositional generalization i.e., that it is not just domain adaptation. I think the submission could do with a better job of dealing with the distinctions between OOD generalization / domain adaptation / concept drift.

- "We propose to obtain compositional representations not from the extractor but reversely from an auxiliary network." At this point, neither the "extractor" nor the "auxiliary network" are defined.

- "These networks can be some existing networks for compositionality learning" If so, what are examples of "existing networks for compositionality learning"?

---

> ### Author Response · Authors · 2020-11-16
> **Reply to Reviewer 2**
>
> Thank you for your feedback.
>
> Q1. The submission claims that other works that investigation compositionality in representation learning do not actually test compositional generalization ("because all combinations have positive joint probabilities in training"). However, I disagree that this is the case in prior work.
>
> A1: Existing methods tend to either measure compositionality but do not measure compositional generalization, or do not use compositional representation. We focus on the case to use compositional representation to achieve compositional generalization.
>
> Q2: The algorithmic components in Section 4 are not adequately motivated, and the relationship of the algorithm to prior work in compositional representation learning is not discussed.
>
> A2: The motivation of the algorithm is in the last two paragraphs of the Introduction section.
>
> Q3: The evaluation tasks are extremely simple.
>
> A3: The compositional generalization works with compositional representation are still in the stage of using simple examples to find the fundamental mechanisms.
>
> Q4: "The main approach for compositional generalization is to learn compositional representations" Is this really the "main approach"?
>
> A4: We will change that this is one approach.
>
> Q5: "We find that the extraction ability does not transfer naturally, because the extraction network suffers from the divergence of distributions" Why is it assumed here that there is an extraction network? The "extraction network" is referred to several times in the introduction and methods section prior to its introduction/explanation.
>
> A5: We focus on the approach with compositional representation, so there is an extraction network.
>
> Q6: "compositional generalization is a type of out-of-distribution (o.o.d.) transferring or generalization, which is also called domain adaptation" This is inconsistent with the previously discussed definition of compositional generalization i.e., that it is not just domain adaptation.
>
> A6: The next sentence after the cited sentence explains the distinction between OOD generalization and compositional generalization: “A sample in such a setting (compositional generalization) is a combination of several components, and the generalization is enabled by recombining the seen components of the unseen combination during inference.”
>
> Q7: "We propose to obtain compositional representations not from the extractor but reversely from an auxiliary network." At this point, neither the "extractor" nor the "auxiliary network" are defined.
>
> A7: We will make them clear.
>
> Q8: "These networks can be some existing networks for compositionality learning" If so, what are examples of "existing networks for compositionality learning"?
>
> A8: Russin et al. 2019 and Li et al. 2019 can be examples, but they are designed for word unit based language tasks as mentioned in the paper.

---

### Decision · Program_Chairs · 2021-01-07
**Final Decision**

**Decision:**

Reject

**Comment:**

This work considers an apparent problem with current approaches to compositional generalisation (CG) in neural networks. The problem seems to be roughly:
1. prior work in CG aims to extract 'compositional representations' from the training distribution
2. work on CG, the training set and the test set are drawn from different distributions
therefore
3. we don't know whether these models can also extract compositional representations from the test distribution

All four expert reviewers were, to differing degrees, confused by this problem framing, largely because they consider the premise (1) to be false.

I am also aware of a large body of recent work on CG in neural networks (see those papers listed by R2) and, as far as i know, none of it involves extracting 'compositional representations' from the training set. Rather, it involves learning something (from the training set) that enables strong performance on a test set that differs from the training set in a way that is informed by ideas of compositionaity.

As far as I know, there are very few  studies that try to identify compositionality by considering the internal representations of neural networks, so it feels incorrect to claim this is standard practice. Any work that goes down this route ought to have a very thorough treatement of the various thorny philosophical and theoretical treatments of compositionality in the literature. As pointed out by R4, the work in its current form does not do this.

In summary, this work attempts to solve a problem that none of the four expert reviewers consider to be in need of a solution.